



# What do the arc magmatism trace element patterns and Sr-Nd-Pb isotopic data reflect: Insight from the Urumieh-Dokhtar magmatic arc of Iran

Mohammad Reza Ghorbani[1], Meysam Akbari[1,2], Ian T. Graham[3], Mathieu Benoit[4], Fatemeh Sepidbar[5]

[1] Department of Geology, Tarbiat Modares University (TMU), Tehran, 14115-175, Iran

[2] Research Institute for Earth Sciences, Tehran, 13185-14194, Iran

[3] Earth and Sustainability Sciences Research Centre, School of Biological, Earth and Environmental Sciences, The University of New South Wales, Sydney, NSW 2052, Australia

[4] Géosciences Environnement Toulouse (GET), Observatoire Midi Pyrénées, Université de Toulouse, CNRS, IRD, 14
Avenue E. Belin, Toulouse, F-31400 France

[5] Department of Geology, Faculty of Science, Ferdowsi University of Mashhad, Mashhad, Iran

*Correspondence to*: Mohammad Reza Ghorbani (ghorbani@modares.ac.ir)

**Abstract.** Mafic volcanic rocks from the Cenozoic Urumieh-Dokhtar Magmatic Arc (UDMA) of Iran, a segment of the Alpine-Himalayan orogenic belt, provide rather restricted ranges of trace element abundances and patterns and Sr-Nd-Pb

isotopic signatures that are distinct enough to help characterize the geochemical signatures inherited from their arc system components. The volcanic rocks are classified into three series; the LILE-rich, LILE-poor, and incompatible trace elements-rich series (ITE-rich series; that include samples with OIB-like patterns). The LILE-rich series is derived from a mantle source metasomatized by wet-slab partial melts whereas the LILE-poor series, high in immobile, highly incompatible elements that include La, Ce, is derived from a mantle source metasomatized by dry-slab partial melts. The ITE-rich series

bear the signatures of mantle metasomatized by slab partial melts that was induced by and reacted with asthenospheric mantle ascended through a slab window or rupture. Given almost primitive geochemical signatures of the mafic rocks, the Sr-Nd isotopic modeling indicate a mantle wedge:slab melt:sediment melt contribution of 45:27.5:27.5 and 55:09:36 for the LILE-rich and LILE-poor series respectively. The mafic volcanic rocks stretching from mantle array (i.e., NHRL) towards enriched mantle on the Pb-Pb isotopic plots support this finding. High Sr and low Nd isotopic ratios of two mafic volcanic

rocks from the LILE-poor series stand in sharp contrast to the other mafic rocks from LILE-poor series that plot near mantle array. Less differentiated signature of these two samples along with their older Nd model ages indicate derivation from older metasomatized mantle segments implying that the source mantle of the UDMA is heterogeneous. Eocene to Early Miocene ages for these three series favor intermittent volcanisms of the three rock series over an extended period of time rather than single episodic magmatism for each geochemically distinct magma series.





## 1 Introduction

The origin and evolution of the subduction-related or arc affinity volcanic rocks have been the subject of scrutiny for many years (Cross and Pilger, 1982; Hildreth and Moorbath, 1988; Benoit et al., 2002; Bouilhol et al., 2013; van Hinsbergen et al., 2020), and those from the Cenozoic Urumieh-Dokhtar Magmatic Arc (UDMA) of Iran are no exception (Berberian and King, 1981; Agard et al., 2005; Chiu et al., 2013; Ghorbani et al., 2014). However, no major review has been undertaken to present an overview of the compositional spectrum and primitive end-members for the magmatic, mainly volcanic rocks from the UDMA. Involvement of a number of distinct source components, such as mantle, fluids or partial melts derived from the subducting slab and overlying sediments, leads to a wide variation in chemistry, particularly trace element abundances and patterns of the subduction-related volcanic rocks (Pearce et al., 1990; Ewart et al., 1998; Xia, 2014). Revealing the source components of volcanic rocks is amongst the most important but also most difficult tasks in the study of magmatic arcs. Moreover, the source components are usually obscured by secondary or differentiation processes. This is especially the case for active continental margins, where the overlying thick crustal rocks may undergo partial melting to further contribute to the magmatism (Spikings et al., 2015; Di Giuseppe et al., 2021). Focusing on the 'most mafic' volcanic rocks, represented by the samples in the 48 to 57 wt. % silica range, is a reliable approach in order to avoid the secondary processes that blur the pristine magmatic components.

Based on the major and trace elements and Sr-Nd-Pb isotopic data of a collection of mafic volcanic rocks sampled from multiple areas across the central part of the UDMA (Fig. 1), this study tries to infer some key implications as to the nature of geodynamic evolution as well as mantle and crustal source regions that melted and /or dehydrated to produce the mafic volcanic rocks. In order to achieve this goal, 1- abundances and patterns of incompatible trace elements and Sr-Nd-Pb isotopic ratios in the mafic rocks are investigated that recognizes three series of mafic rocks and 2- considering the likely involvement of the mantle, slab fluids, slab melts and crustal components as well as the radiometric ages compiled from recent studies, a petrogenetic model is presented that sheds light on the geodynamic evolution of the three rock series introduced.

## 2 Geology of the study area

As a part of the Alpine-Himalayan Belt, the Zagros Orogeny stretches for nearly 2000 km from northwest to southern Iran. This Orogeny is the consequence of subduction of the Neotethyan oceanic plate beneath central Iran and the final collision of the Arabia and Eurasia plates (Berberian and King, 1981; Alavi, 1994; Mohajjel and Fergusson, 2000). The Zagros Orogeny is composed of three sections, the Zagros Fold and Thrust Belt (ZFTB), the Sanandaj-Sirjan Zone (SSZ), and the UDMA (Fig. 1.a). The UDMA, also known as the central Iran magmatic belt, includes a thick mainly Eocene to Miocene volcanic succession of up to 3000 m that is intruded by plutonic bodies of mainly Oligocene to Miocene ages. An 'active continental margin' model (Berberian and Berberian 1981; Moinevaziri 1985) for the subduction predominates though a few studies report some similarities to 'island arc' magmatism (Ghorbani, 2006; Shahabpour, 2007).





**Figure 1: (a) Regional tectonomagmatic units of Iran (after Alavi, 1994). The gray fields show the Urumieh-Dokhtar magmatic arc (UDMA). Abbreviations: SSZ, Sanandaj-Sirjan Zone; ZFTB, Zagros Fold and Thrust Belt. (b) Simplified geological map of the central UDMA which is taken from the Geological maps 1:100,000 published by Geological Survey of Iran. More detailed geological maps are provided for three segments of the central UDMA (see "Geology of the study area" section for references). Location of the samples selected for geochemical analyses are also shown. For legend and lithostratigraphic columns of these three segments of the central UDMA see Fig. S1.**

Based on the interbedded fossiliferous beds, the UDMA volcanic succession has been traditionally attributed to the Eocene (e.g., Berberian and King, 1981; Emami et al., 1993). Verdel et al. (2011) based on the dominantly Eocene ages obtained for the volcanic rocks from the Tafresh area (i.e., the northernmost part of the central UDMA), introduced the Eocene flare-up model as being responsible for the Cenozoic magmatism of the UDMA whereby extension induced partial melting of the metasomatized mantle that produced the Eocene calc-alkaline volcanic rocks. This model suggested that the gradual slab rollback caused asthenospheric upwelling that triggered the Oligocene OIB affinity volcanic rocks.

Multiple age dating of the volcanic rocks from the UDMA prompted Chiu et al. (2013) to infer that magmatic flare-up was not limited to the Eocene but also extended into the Oligocene time. Further age dating achieved by Ghorbani et al. (2014) extended the widespread magmatic activity in the UDMA even further into the Miocene. More recently, Moradi et al. (2021) also reported the Miocene ages for the volcanic rocks from the Neragh area of the central part of the UDMA. Adakitic rocks have also been reported from the central UDMA (Omrani et al., 2008; Ghorbani and Bezenjani, 2011; Ghorbani et al., 2014; Ahmadian et al., 2016); these however postdate the Middle Miocene and have been attributed to high-pressure slab melting.

The present literature on the evaluation of the compositional aspects of the magmatic rocks from the UDMA is limited. Volcanic rocks with negative Nb-Ta anomalies are attributed to subduction (Pearce, 1982; Kelemen et al. 2005) whereas some rocks with higher Nb-Ta abundances have been regarded as being derived from an asthenospheric mantle source (Verdel et al., 2011; Yeganehfar et al., 2013). A few studies carried out in the UDMA include Sr-Nd-Pb isotopic data on the mafic volcanic rocks (Yeganehfar et al., 2013; Jolani Varzeghani, 2017; Khodami, 2019; Sepidbar et al., 2019; Moradi et al., 2022); these works collectively attribute the volcanic rocks to the mixing of varying, marginal proportions of crustal melts with dominating mantle-derived partial melts in a subduction setting. Scarcity of the mafic rocks in the UDMA volcanic successions has been a major obstacle in the evaluation of end-members of the source regions and materials that evolved to generate the volcanic successions. The paucity of mafic rocks makes it difficult to follow the melt compositional evolution or liquid line of descent (Wilson, 1989). To overcome the paucity of mafic rocks, the current study carried-out widespread sampling from multiple areas across the central part of the UDMA (Fig. 1). This furnished a representative set of the mafic-intermediate volcanic rocks for detailed petrological and geochemical investigations.

The volcanic rocks sampled in this study are lavas ranging in thickness from meters to tens of meters. Porphyritic to aphyric textures of the volcanic rocks confirm their eruptive nature. A simplified geological map of the UDMA is shown in Fig.1, which is compiled based on the geological maps on scale 1:100.000 of Kahak (Ghalamghash et al., 1988), Kashan (Radfar





and Alaie-Mahabadi, 1993), Natanz (Khalatbari Jafari and Alaie-Mahabadi, 1995), Tarq (Zahedi and Rahmati, 2002), Ardestan (Radfar and Amini Chehragh, 1999), Shahrab (Bahroudi and Fonoudi, 1997), Kuhpayeh (Radfar et al., 2002), 100  Kajan (Amini and Amini Chehragh, 2003), Nain (Alaie-Mahabadi and Foudazi, 2004), Sarve-Bala (Amidi et al., 1989), and Kafeh-Taghestan (Ghalamghash et al., 2005). The present study area comprises three segments from northwest to southeast; these are Kahak (including Fordou and Vadghan), Ardestan (including Mishab, Marbin, and Kahang), and Nodoushan, respectively (Fig. 1.b). Integrating these volcanic segments into a larger tectonic context will be a worthwhile endeavor as a roadmap for future studies.

## 3 E$_1$-E$_6$ subdivisions versus age dated volcanic succession

The UDMA is mainly dominated by intermediate to felsic volcanic and pyroclastic rocks, which are outcropped in the Cenozoic orogen. These include andesites-trachyandesites, dacites-rhyolites, and tuffs-tuff breccia-ignimbrites respectively. Mafic volcanic rocks occur sporadically and include basalts, alkali basalts, and trachybasalts. The volcanic successions that constitute the UDMA were divided into six units, E$_1$-E$_6$, on the geological maps drafted by the Geological Survey of Iran 110  (c.f., for lithostratigraphic columns of different segments of the study area see Fig. S1). Having presented the age-dated samples from central UDMA on a generalized stratigraphic column that was compiled based on the geological maps, Ghorbani et al. (2014) implied a discrepancy. The discrepancy in ages (i.e., the difference between the real age and the presumed Eocene age), indicates the necessity for a revision in the E$_1$-E$_6$ divisions. An improvement in the stratigraphic column of the UDMA has recently begun to emerge (see Fig. 2 in: Sepidbar et al., 2021); that is the Oligocene-Miocene 115  magmatic records grow at the expense of Eocene magmatic records.

## 4 Petrography analysis

Petrography is focused on samples of the volcanic series considered representative of the present study areas from the UDMA (Fig. 1.b). The basaltic rocks are intended for the present study; these included pyroxenes as the phenocrysts and/or in the groundmass. Most of the samples contain a few plagioclase and clinopyroxene phenocrysts. The mafic–intermediate 120  volcanic rocks from the study area are slightly to moderately porphyritic in texture (Fig. 2). While the groundmass in some samples is almost aphyric, other samples show a microcrystalline groundmass composed of plagioclase, clinopyroxene, Fe-Ti oxides, and glass.

Typically, samples with a microcrystalline groundmass contain a few pseudomorphs after olivine (Figs. 2.a-d) whereas samples with an aphyric groundmass include plagioclase and a few pseudomorphs after orthopyroxenes (Figs. 2.e-f) or a few 125  fresh orthopyroxenes (Figs. 2.g-h). Glomeroporphyritic textures are also present and often include clinopyroxene, plagioclase, and pseudomorphs after olivine or orthopyroxene. Only fresh samples are selected, so secondary minerals such as sericite, chlorite, calcite and epidote which are the alteration products of silicate minerals are rare if present at all. Before





using geochemical data, however the possibility of alteration and its effect on element mobility are discussed below (see section 7.1).

130





**Figure 2: Photomicrographs of the major mafic volcanic rock types from the Central UDMA in thin section, in plane-polarized light on the right, and the same fields of view in cross-polarized light on the left. Field widths are all 4 mm. Abbreviations: Ol, olivine; Opx, orthopyroxene; Cpx, clinopyroxene; Pl, plagioclase.**

## 5 Methods

A representative set of 66 mafic–intermediate volcanic rocks from the study area was analyzed for major and trace elements (Table S1). After a detailed examination of the major and trace element geochemistry of the rocks, 14 samples were selected for Sr-Nd-Pb isotopic analyses. In order to avoid altered samples, the representative set was selected based on detailed petrographic examination of 250 thin sections across the region.

### 5.1 Major and trace element analysis

Whole-rock analyses for major elements were obtained by X-ray fluorescence (XRF) at the University of New South Wales, Sydney, Australia, following the procedures of Norrish and Hutton (1969). The analyses were carried-out on a Phillips PW2400 XRF spectrometer using 40-mm glass disks. Inductively coupled plasma–mass spectrometry (ICP-MS) was used for rare earth and trace elements analyses of the whole-rock samples at the Genalysis Laboratory, Perth, Australia, using the Perkin–Elmer ICP-MS 9000 (i.e., 51 samples marked by numbers in normal fonts in Table S1) and the Melbourne University, Victoria, Australia using the NexION 2000 (i.e., for 15 samples marked by numbers in italic fonts in Table S1). Accessory minerals are trace element repositories that resist the mixed acid digest method. To ensure complete digestion while maintaining good sensitivity, a combination digest method (i.e. a mixed acid digest and multi-fusion) was used. The details are as follows. The pulp was digested in $HNO_3/HClO_4$ on a hot plate and then cooled. HF was added and digestion continued on the hotplate; the mixture was then cooled and salts were leached with an $HCl/HNO_3$ mix on the hotplate. The solution was then cooled and filtered to recover undigested material fused with minimal Li metaborate–tetraborate flux. The melt was produced and then leached with HCl. Leach solutions were combined for the ICP-MS analysis.

In-house standards were analyzed to check the accuracy of the methods. Analytical reproducibility was also examined. For the rare earth and trace elements, errors were evaluated to be better than 3% except for Nb, Ta, and U (8%). Duplicate analyses indicate that the errors for major elements are better than 1% except for alkaline elements (5%). The whole-rock analysis totals are in the range from 98.79 to 101.83 wt% with a mean of 100.80 wt% (Table S1). The diagrams depicted are based on major element analysis recalculation to 100 wt%, on an anhydrous basis.

### 5.2 Sr-Nd-Pb isotopic analysis

The isotopic data was obtained in GET-OMP, Toulouse University (France), using the Thermo Scientific TRITON+ solid source mass spectrometer, following Labou et al. (2010) and Li et al. (2011, 2012) procedures. Before measurement, about





100 mg of whole rock powder was weighed in a teflon beaker and dissolved in a mixture HF/HNO$_3$ 1:1. After dissolution, samples were diluted in 1 ml, 2% HNO$_3$ and Nd/Sr were extracted from the matrix (2N HNO$_3$) using a combination of Sr-Spec and Thru-spec Eichrom resins. Mixed Sr and REE were loaded on a Re filament and were run sequentially (first Sr then Nd) using a double Re filament protocol. Monitoring of the interferences of $^{87}$Rb and $^{144}$Sm were proceeded according to the protocol of Li et al. (2012) and the quality and reproducibility of the measurements were controlled using a sequential measurement of isotopic standards (SRM 987 and JNdi), doped isotopic standards (NBS 987+ Rb and JNdi + Sm) and laboratory-dedicated Sr+REE artificial solutions. Standard reproducibilities' are $^{87}$Sr/$^{86}$Sr = 0.710280+/-15 for SRM-987 and 0.512100 +/- 10 for JNdi (pure and doped) and fall within the recommended values. Measured blanks are 25 pg for Nd and 325 pg for Sr. $^{87}$Sr/$^{86}$Sr and $^{143}$Nd/$^{144}$Nd ratios were normalized against $^{86}$Sr/$^{88}$Sr = 0.1194 and $^{146}$Nd/$^{144}$Nd = 0.7219 respectively, after corrections from isobaric interferences using $^{87}$Rb/$^{85}$Sr = 0.387041 on $^{87}$Sr and combination of $^{147}$Sm/$^{149}$Sm=1.08583 and $^{147}$Sm/$^{144}$Sm = 4.87090 on $^{144}$Nd.

Pb fractions have been extracted from the matrix during the same elution protocol as Sr and Nd following Pin and Gannoun (2017) procedure. Pb isotopes have been determined on a Thermo Scientific TRITON+ solid source mass spectrometer using Rex Taylor Double Spike method.  NBS 981 reproducibility on lead ratios is around 100 ppm. Blanks are typically lower than 20 pg.

## 6 Results

### 6.1 Major and trace elements

The mafic volcanic rocks total vary in the range 98.28 to 101.82 wt% with an average of 100.46 wt%. Most samples contain less than 3 wt% LOI (loss on ignition), an indication of the reliability of their geochemistry (i.e., the rocks have maintained their original geochemical characteristics, see the subtitle Alteration in Discussion section). On Harker variation diagrams, CaO, MgO and FeO show broadly decreasing trends with increasing SiO$_2$ whereas Na$_2$O and K$_2$O increase (Fig. 3). These covariations are well defined for specific areas (c.f., Figs. 3 and S2a, b). Volcanic rocks from Marbin and Kahang show higher Mg and Ca and lower Fe, Na, and K as compared to the Fordou and Vadghan areas (c.f., Figs. S2a, b). These largely overlap volcanic rocks from the LILE-poor and LILE-rich series (see below) respectively.

The mafic volcanic rocks from the study area can be divided into three distinct compositional series based on their primitive mantle (Sun and McDonough, 1989) normalized trace element patterns (on Fig. 4 these are termed the LILE-poor, the LILE-rich, and the ITE-rich series (LILE and ITE refer to large-ion lithophile elements and incompatible trace elements) respectively. The LILE-poor and LILE-rich series show negative Nb-Ta anomalies, which clearly favour an arc setting. These rocks belong to the subalkaline series and further show calc-alkaline affinities. The LILE-poor series is rather depleted in Rb, K, and HREE compared to LILE-rich series. The ITE-rich series is enriched in the lithophiles, La, Ce, Pb, Sr and P with both calc-alkaline and alkaline features. Mafic volcanic rocks from the ITE-rich series are sometimes enriched in Nb-Ta (Fig. 4; diagrams in the upper row).





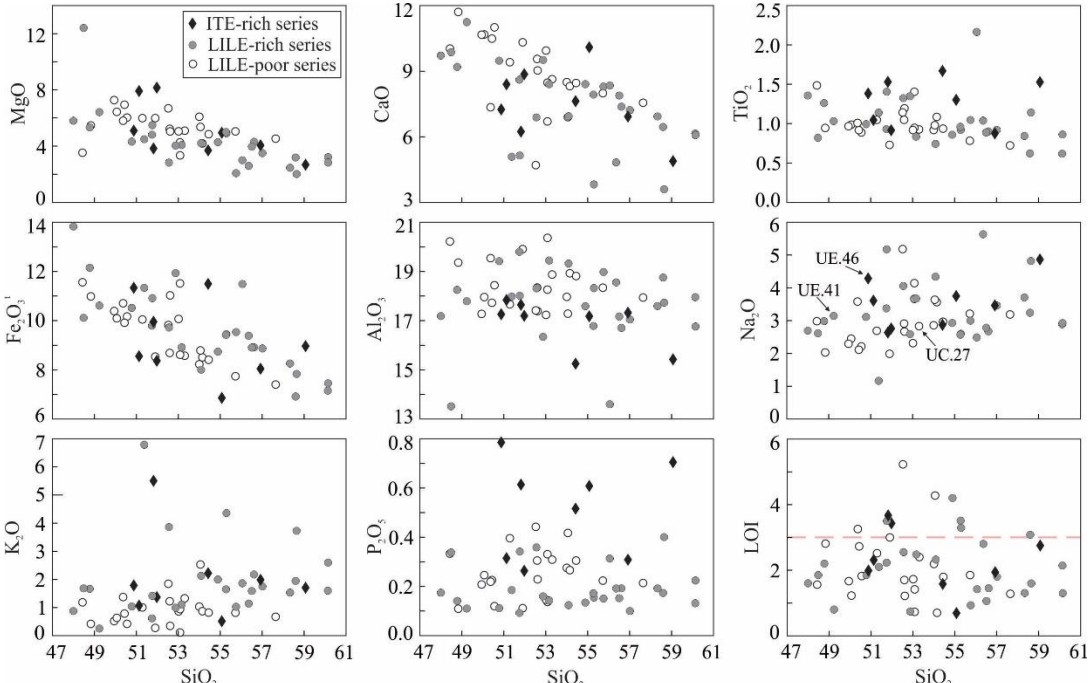


**Figure 3: Major element oxides vs. SiO₂ variation diagrams for the LILE-poor, LILE-rich, and ITE-rich series from the UDMA. For details of the variation diagrams also see Figs. S2.a-c.**

Volcanic rocks presenting these three patterns (i.e., the LILE-poor, LILE-rich and ITE-rich series) show some major element

geochemical distinctions as well. Rocks from the LILE-poor series show higher Mg and Ca and lower Fe, and alkaline elements as compared to the LILE-rich series samples (Figs. 3 and S2.a-c). Details of the mafic rocks variation diagrams for three segments of the study area are shown in Fig. S2.a-c; these are Kahak (Fodou and Vadghan), Ardestan (Mishab, Marbin and Kahang), and Nodoushan (Fig. 1). Importantly, the mafic volcanic rocks from the three series/patterns occur across all segments of the study areas, though vary in their different proportions.


### 6.2 Sr-Nd-Pb isotopic data

Sr-Nd-Pb isotopic ratios were determined for a set of fourteen mafic volcanic rocks (Table S2) from the study areas belonging to the LILE-rich series (four samples), the LILE-poor series (six samples) and the ITE-rich series (four samples). The ranges of measured Sr isotopic ratios for these three series are 0.70562 to 0.706085, 0.705099 to 0.706263, and

0.704443 to 0.705561 respectively. The ranges of measured Nd isotopic ratios for these three series are, 0.512715 to 0.512786, 0.512607 to 0.512766, and 0.512673 to 0.512804 respectively.







**Figure 4: Primitive mantle normalized trace element patterns for the Kahak (including Fordou and Vadghan), Ardestan**
**(including Mishab, Marbin, and Kahang), and Nodoushan mafic volcanic rock. These are classified into the LILE-rich, LILE-poor, and ITE-rich series. Differentiated samples are shown in pale-colored as compared to the more primitive samples which are shown in dark colors. Underlines sample numbers indicate the mafic volcanic rocks for which Sr-Nd-Pb isotopic data are obtained in the current study (see Table S2 and Fig. 5). Age dated samples are from Ghorbani et al. (2014). For other age dated samples from the study areas see Figs. S3 and S4. Primitive mantle (PM) normalization values and trace element abundances for the OIB**
**and P-MORB (diagrams in the upper row) are from Sun and McDonough (1989). Trace element abundances for the island arc basalts (lower right diagram) and active continental margin basalts (middle left diagram) are from Jicha and Singer (2006) and Straub et al. (2015) respectively.**

Only four out of the fourteen samples are age dated by whole rock K-Ar method (Ghorbani et al., 2014). These range from
18.2 to 34.6 Ma similar to the ages obtained for the UDMA mafic volcanic rocks by Yeganehfar et al. (2013). None of the three series of volcanic rocks (the LILE-rich series, the LILE-poor series and the ITE-rich series) seems confined to a particular time span. Mafic volcanic rocks of the same compositional affinities from the central UDMA studied by Moradi et al. (2022) further extended this age range and show ages of 60.0 to 21.5 Ma. Therefore, an average age of 35 Ma is suggested for calculation of the initial Sr, Nd and Pb isotopic ratios. The ranges of initial Sr isotopic ratios for these three
series are, 0.705448 to 0.706017, 0.705044 to 0.706237 and, 0.704379 to 0.705505 respectively. The ranges of initial Nd isotopic ratios for these three series are 0.512677 to 0.512746, 0.512573 to 0.512729 and 0.512637 to 0.512772 respectively. (Table S2).

Mafic volcanic rocks from the study area plot on the same area of the Sr-Nd isotopic plot as other orogenic volcanic rocks and on the same area as other mafic volcanic rocks from the UDMA (Fig. 5). Toward lower Sr isotopic ratios, data points
from the three series converge to share a narrow space in the mantle array close to the BSE. The Sr-Nd isotopic variation trends expand towards higher Sr and lower Nd isotopic ratios (Fig. 5.a). These volcanic rocks display moderately radiogenic Pb compositions ($^{206}Pb/^{204}Pb(i)$ = 18.52-18.72, $^{207}Pb/^{204}Pb(i)$ =15.51-15.69, and $^{208}Pb/^{204}Pb(i)$ = 38.33-38.80; Table S2). On Pb-Pb isotopic plots the mafic volcanic rocks form linear trends stretching from mantle array (i.e., NHRL) towards enriched mantle, that is above the reference line (Fig. 5.b, c).

**7 Discussion**

**7.1 Alteration**

The development of secondary minerals in the mafic volcanic rocks from the study areas is strictly limited to a few pseudomorphs after primitive olivine and orthopyroxene (see petrography). This along with LOI values lower than 3 wt. % in most of the samples (i.e., in 55 out of 66 samples) indicates negligible alteration effects. In addition, most of the samples
from the LILE-poor series have the same LOI abundances as the samples from the LILE-rich series (Figs. 3 and S2.a-b). These overlapping volatile contents do not support a correlation between the LOI and alkali contents (see Fig. 3) which is



usually regarded as a measure of the extent of alteration (e.g., Dong et al., 2017). In Mishab, the LILE-poor series samples have the highest LOI; this does not support a correlation between alkali content and LOI and provides yet additional evidence as to the lack of significant alteration effects.


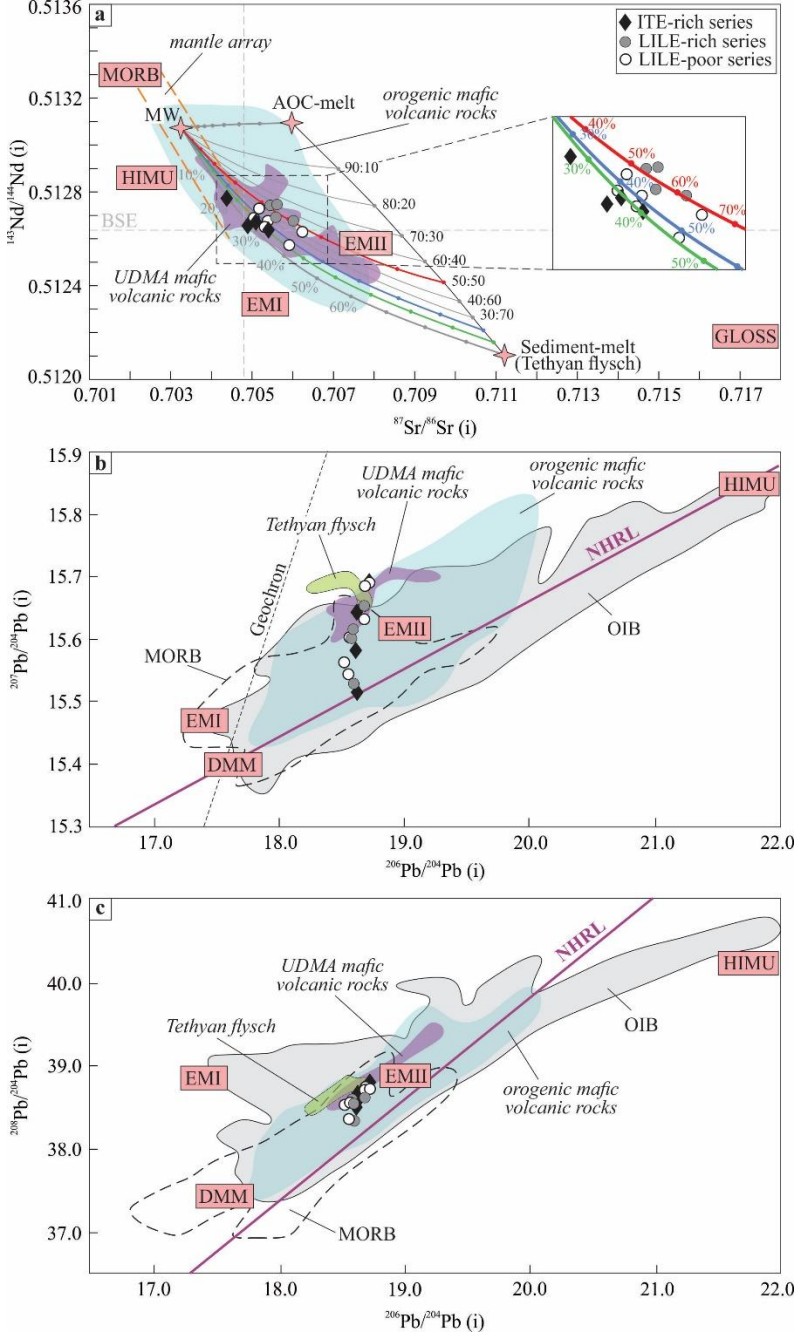





**Figure 5: (a) Initial Sr-Nd isotopic ratios plot for the mafic volcanic rocks from the study areas, showing a three-component mixing model for the LILE-rich, LILE-poor, and ITE-rich series mafic volcanic rocks from the central UDMA. See Table S3 for further details on modelling parameters. (b) $^{207}Pb/^{204}Pb$ vs. $^{206}Pb/^{204}Pb$ and (c) $^{208}Pb/^{204}Pb$ vs. $^{206}Pb/^{204}Pb$ ratios plots for the mafic**
**volcanic rocks from the study areas. Average values for DMM (depleted MORB mantle), HIMU (high μ; μ = $^{238}U/^{204}Pb$), EMI (enriched mantle 1), and EMII (enriched mantle 2) are calculated based on the samples presented in Akbari et al. (2023) that include a comprehensive database of a variety of basalts. GLOSS (global subducted sediment) is adapted from Plank and Langmuir (1998). The fields for OIB (oceanic island basalts) and MORB (mid-ocean ridge basalt) are from Stracke et al. (2003, 2005), Chauvel et al. (2008), and references therein. NHRL (Northern Hemisphere Reference Line) is adapted from Hart (1984).**
**The mantle wedge is represented by the average of Kamchatkan arc volcanic rocks (Kepezhinskas et al., 1997), the altered oceanic crust (AOC) is adopted from Hauff et al. (2003), and the Tethyan flysch sandstone sample 06FL03 from Serbia is adopted from Prelević et al. (2008). For comparison of the data achieved in the present study with published isotopic data for the W Nain (Yeganehfar et al., 2013) and Nodoushan (Jolani Varzeghani, 2017) areas see Fig. A5. The field for orogenic mafic volcanic rocks is adapted from GEOROC database. The distribution of nearly coeval mafic volcanic rocks from UDMA (Omrani et al., 2008;**
**Yeganehfar et al., 2013; Shafaii-Moghadam et al., 2014; Jolani Varzeghani, 2017; Khodami, 2019; Rabei et al., 2020; Moradi et al., 2021) are also shown for comparison. See text for more details.**

## 7.2 Primitive signatures

Mafic magmatic rocks are not common in magmatic arcs (e.g., Plank and Langmuir, 1988). Subduction or arc-related
volcanic successions are dominated by the intermediate to felsic volcanic products and these are conventionally viewed as derivatives of arc basaltic magmas which have been triggered by partial melting of metasomatic components in the mantle wedge (Gill, 1981; Plank and Langmuir, 1988; Benoit et al., 2002; Streck et al., 2007). A thick continental lithosphere and/or crust promotes the development of high-level magma chambers that prohibits rapid magma ascent and leads to significant magmatic differentiation, namely Assimilation Fractional Crystallization (AFC), and this masks most of the original source
mantle signatures (Plank and Langmuir, 1988; Farner and Lee, 2017). MgO contents of the volcanic rocks from the study area are rather low (< 7 wt.%) indicating that even the most mafic samples experienced some degrees of differentiation. It is however noted that some of the least differentiated samples from the magmatic arcs maintain the trace element signatures of primitive melts. Non-cumulative, rather aphyric or slightly porphyritic UDMA mafic rocks with low LOI contents and minor or no alteration, are examined here to infer implications as to the geochemical signatures of their primitive melts.

It is noted above that the volcanic rocks from the study area are representative of three distinct trace element patterns, namely the LILE-poor, LILE-rich, and ITE-rich series. Samples from the LILE-poor and LILE-rich series are marked by negative Nb-Ta anomaly characteristics of the subduction related settings (Fig. 4; see lower right and middle left diagrams for island arc and active continental margins volcanic rocks trace element patterns respectively). The LILE-poor series are depleted in the lithophiles whereas the LILE-rich series are enriched in the lithophiles. The trace element pattern for the ITE-





rich series is significantly different; it is enriched in the lithophiles, La, Ce, Pb, Sr, and P. Samples from the ITE-rich series are rather enriched in Nb-Ta and present OIB-like trace element patterns (Fig. 4; diagrams in the upper row).

The LILE-poor and LILE-rich series trace element patterns span a wide spectrum from an overall low trace element abundance to an overall high/elevated trace element abundance. The patterns with an overall low trace element abundance (i.e., of < 10 times primitive mantle) show less differentiated signatures (e.g., slight or no negative Sr and Ti anomalies) and

are suggested to represent more primitive partial melts. The low trace element abundance patterns are likely to imply high-degree partial melts (Wilson, 1989). Higher-degree partial melts show lower trace element abundances as well as evidence supporting the exhaustion of fusible mantle mineral phases (Ghorbani and Middlemost, 2000). A pronounced negative anomaly in Zr-Hf in some of the UDMA mafic volcanic rocks with an overall low trace element abundance pattern (Fig. 4) appears to indicate the significant contribution of a mineral phase that bears a negative Zr-Hf anomaly. The gradual

consumption and final disappearance of clinopyroxene in the mantle source, as the major participating phase, could impose its low Zr-Hf signature (Linnen and Keppler, 2002; Atanasova et al., 2020) on the partial melt produced. Alternatively, it is likely to be an indication of metasomatism by slab partial melts in the source mantle of the LILE-poor series. Metasomatism by slab partial melts is associated with the development of Al-rich orthopyroxene and Mg-rich amphibole in the mantle (Schiano et al., 1995; Kepezhinskas et al., 1995; Szabó et al., 2004). This contracts the clinopyroxene (Cpx) stability field so

that the LILE-poor series parental melts portray signatures of Cpx exhaustion. Orthopyroxene megacrysts (i.e., as pseudomorphs) in one of the samples from the LILE-poor series (Fig. 2) support this hypothesis. These may be the remnants of the disaggregation of such metasomatized mantle xenoliths.

**7.3 Does fractional crystallization provide an interseries link?**

The fact that two series of the samples with the most primitive geochemical signatures (i.e., from the LILE-poor and LILE-

rich series) show significant differences in LILE implies that the differences/distinctions are source inherited. In other words, more primitive samples from these two series best represent their compositional distinctions. Hence, the LILE-poor and LILE-rich series are likely rooted in different source regions. One might challenge this hypothesis and assign the geochemical differences between the two series to differentiation from a single parental melt. Fractionation of the liquidus phases crystallizing from primary mafic melts (i.e., Ol, Opx, Cpx, Pl; a single phase or a mineral assemblage) may explain

some geochemical differences between the two series. Fractionation of Ol + Pl ± Cpx from an assumed primary mafic melt can explain the major element geochemical evolution from the LILE-poor series to the LILE-rich series rocks (i.e., decreasing melt Mg and Ca, and increasing Fe and alkaline elements). However, this model is not able to explain the distinction/differences between the trace element abundances and patterns of the two series. Such a differentiation model is expected to produce a series of parallel trace element patterns wherein the abundances of nearly all incompatible trace

elements (i.e., with the exception of P and Sr due to the fractionation of apatite and plagioclase, respectively) increase as differentiation proceeds. Moreover, the differentiation model is not able to explain some other issues as follows: 1- negative Th anomalies of the LILE-rich series samples are not consistent with their more differentiated nature. Th is a highly





incompatible element, and is expected to increase with differentiation. Th can be compatible in apatite, however rather constant, minimal $P_2O_5$ abundances in the LILE-rich series do not support apatite fractionation. 2- the volcanic rocks
representing the LILE-poor and LILE-rich series stretch parallel to each other across the compositional spectrum at the same silica range (Fig. 3). This is more consistent with the concurrent evolution of the two series rather than one being the differentiation product of the other.

As noted above, the LILE-poor and LILE-rich interseries geochemical differences are unlikely to have been interconnected (i.e., the latter is not developed by differentiation of the former). Rather, the two series bear some primitive signatures and
are likely derived from two distinct source mantle regions. Nevertheless, the LILE-poor and LILE-rich intraseries geochemical variations appear to have been developed by differentiation through fractional crystallization (FC). The intraseries FC model is supported by the following four observations.1- The presence of the Ol, Opx, Cpx, Pl, and Fe-Ti oxides phenocrysts in the UDMA volcanic rocks. 2- The development of elevated, parallel trace element patterns in the more evolved samples of the respective series (i.e., the LILE-poor and LILE-rich series; Fig. 4). 3- The gradual decrease of some
trace elements which have high Kd for the fractionated minerals (e.g., development of negative Ti anomaly) in the more evolved samples. 4- The plots of differentiation index versus the Sr and Nd isotopic ratios confirm the dominance of FC with little or no crustal contamination in evolution of the volcanic rock series from the study area (Fig. 6; for details on the one sample from ITE-rich series and the two samples from the LILE-poor series that are marked with arrow and show different geochemistry as compared to the mainstream respective series see Sections 7.5 and 8). Pb isotopic ratios as more sensitive
tool for detection of crustal components however, indicate some crustal contamination. Decreasing $^{207}Pb/^{204}Pb$ with increasing silica (Fig. 6) for most samples indicates interaction with unradiogenic lower crust.

Volcanic rocks with the most primitive geochemical signatures are presented in the current study, however Nodoushan rocks contain >54% silica and are rather differentiated. This is indicated by Nodoushan rocks pronounced negative anomalies for Ti and P as well as their elevated trace element patterns (Fig. 4). In fact, Nodoushan rocks constitute two sets of
compositional variations, samples with stronger negative anomaly for Sr (i.e., the more differentiated set) are richer in incompatible trace elements as compared to the more primitive set. In Fig. 4 these are shown by green and blue patterns respectively. The more differentiated set show lower Al abundances attesting to the higher plagioclase differentiation (Fig. S2.b).

### 7.3 Mantle metasomatism by slab fluids and slab partial melts; furnishing sources of the LILE-rich and LILE-poor
**series respectively**

Different modes of trace element enrichment (i.e., by fluids/melts released from the subducting slab) in the source mantle might be responsible for the development of the LILE-poor and LILE-rich series melts. Melting of the mantle metasomatized by wet-slab-melt is suggested to have produced the LILE-rich series melts whereas melting of the mantle metasomatized by the partial melt of the same but rather dehydrated slab segment (dry-slab-melt), is likely to have led to the LILE-poor series



melts. This is consistent with the Mg-richer composition of the latter (Fig. 3). This occurs because the mantle with lower

lithophiles and volatiles, has a higher solidus and is more magnesian in composition (Pickering-Witter and Johnston, 2000).

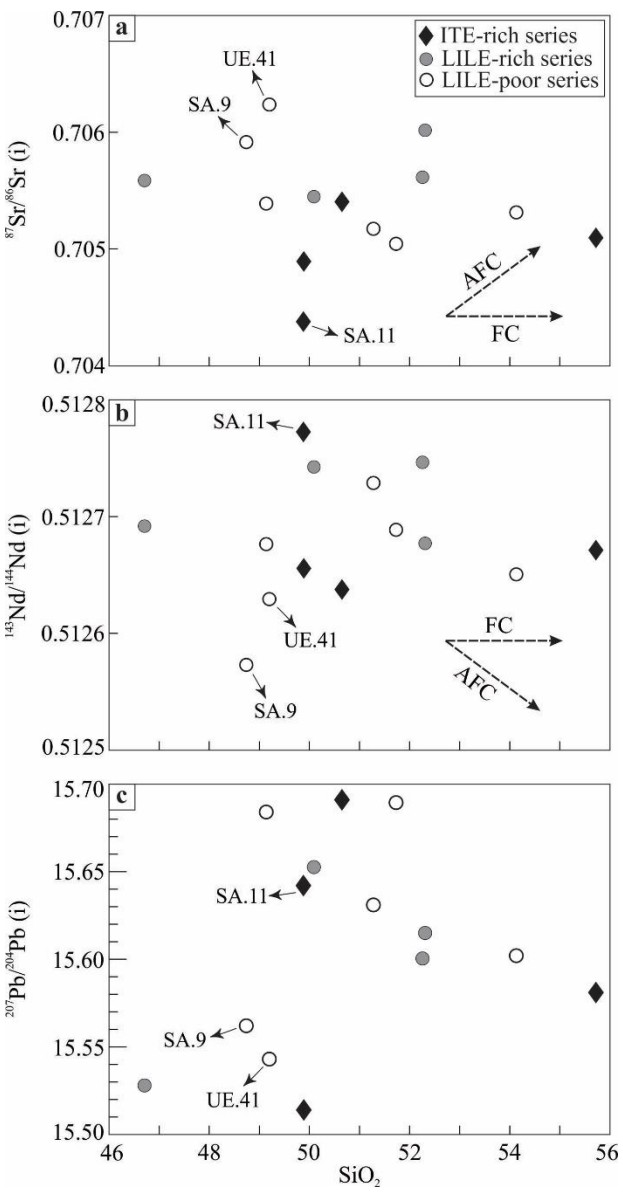

**Figure 6: Plots of (a) $^{143}Nd/^{144}Nd(i)$, (b) $^{87}Sr/^{86}Sr(i)$ and (c) $^{207}Pb/^{204}Pb(i)$ vs. SiO₂, as a fractionation index, for the LILE-rich, LILE-**

**poor, and ITE-rich series mafic volcanic rocks from the central UDMA; details in the text.**

The contribution of higher fluids in evolution of the LILE-rich series is implied by the trace element ratio plot (Fig. 7). The

petrography is also in line with this model (i.e., wet-slab-melt versus dry-slab-melt involvement in the petrogenesis of the





LILE-rich and LILE-poor series respectively). Higher mobile incompatible element abundances of the primitive melt in the

LILE-rich series, are responsible for its lower melt viscosity that is reflected in the aphyric groundmass of samples from this series (Fig. 2). In contrast, the microcrystalline groundmass of the samples from the LILE-poor series is due to the higher viscosity of their primitive melts as a consequence of their mantle source metasomatism by silica-rich slab partial melts. Two different scenarios are raised here that might account for the petrogenesis of these two major rock series.

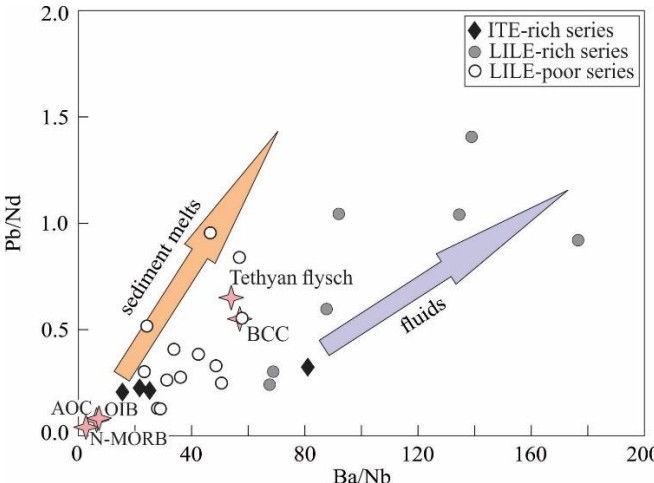


**Figure 7: Ba/Nb versus Pb/Nd plot for the mafic volcanic rocks (SiO₂ < 52 wt%) from the study area. Higher mobile/immobile trace elements ratios in the LILE-rich series indicate the involvement of higher fluid in petrogenesis of this series (e.g., Duggen et al., 2007; Straub et al., 2015). For references of the components involved (OIB, N-MORB, AOC, Tethyan flysch) see the caption for Fig. 5.**


### 7.3.1 The slab roll-back scenario

Considering the emphasis has recently been made in regard with the application of slab roll-back model in petrogenesis of the Cenozoic magmatic arc from Iran (Agard et al., 2005; Verdel et al., 2011; Yeganehfar et a., 2013: Babazadeh et al., 2017: Moradi et al., 2021), one might prompt a two-stage mantle metasomatism as responsible for petrogenesis of the two series of

volcanic rocks from the study area as follows. Slab dehydration at a given depth produced the 'lithophile enriched' source mantle that generated the LILE-rich series melts. Gradual deepening (or roll-back) of the dehydrated, lithophile-depleted slab, was associated with an increasing temperature that prompted partial melting of the slab. This slab melt was depleted in lithophiles but enriched the overlying mantle in 'immobile, highly incompatible elements'. Due to the involvement of slab partial melts, the LILE-poor series melts are enriched in the immobile trace elements Nb to P (i.e., on the normalized trace

element patterns; Fig. 4) as compared to the LILE-rich series. This scenario however, does not stand close scrutiny. A review of the age dated mafic rocks from the study areas (Ghorbani et al., 2014; see Fig. 4) and the areas nearby (Ghorbani and





Bezenjani, 2011; Yeganehfar et al., 2013; see Figs. S3 and S4) do not indicate that the LILE-poor series postdate the LILE-rich series.

### 7.3.2 Simultaneous slab fluids and slab melts mantle metasomatism scenario

It is suggested here that the wet-slab-melt and the dry-slab-melt metasomatism, leading to the development of the LILE-rich and the LILE-poor series, took place in a short time span. At first, hydrous slab melting at a given depth lead to the mantle metasomatism by hydrous slab partial melts. It is shortly followed by a modest change in the physiochemical conditions that governed the slab-mantle wedge system. The new conditions prompted partial melting of the slab that has already undergone dehydration, leading to the development of second type of mantle metasomatism by anhydrous slab partial melts.

### 390    7.4 The incompatible trace elements-rich (ITE-rich) series (including OIB-like samples)

    Samples from the ITE-rich series are enriched in a wide range of incompatible trace elements that include lithophiles, La, Ce, Pb, Sr, and P, so they are called the incompatible trace elements-rich series, ITE-rich series (Fig. 4). These include the most Nb-Ta enriched samples from the study area that approach the OIB pattern, therefore might be called the OIB-like samples. However, they are not genuine OIB as; 1) their trace element patterns are rather differentiated (i.e., as compared to

the smooth trace patterns of the OIBs; Fig. 4), 2) their trace element abundances are not as high as OIBs, and 3- their Sr-Nd-Pb isotopic signatures partly overlap those of the LILE-rich and LILE-poor series (Fig. 5), however Sr and Nd isotopic ratios for the ITE-rich series extend toward lower and higher values respectively (i.e., as compared to the LILE-rich and LILE-poor series).

    Samples from the ITE-rich series are more like the high-Nb basalts (HNBs; Defant et al., 1992). The occurrence of HNBs in

the subduction frameworks is attributed to either a mantle plume (i.e., an asthenospheric or OIB-type mantle) or a metasomatized mantle wedge (Reagan and Gill, 1989; Castillo et al., 2007). The HNBs are also known to be the mixing products of an enriched (e.g., OIB-type) and a depleted (e.g., arc-type) source mantle (Hastie et al., 2011; Ahmadvand et al., 2021; Akbari et al., 2022). The ITE-rich series is suggested to have been generated by slab-window or slab-tearing process at moderate pressures whereby the underlying asthenospheric mantle penetrated in and facilitated slab partial melting that

interacted with and metasomatized mantle to produce the ITE-rich series (Benoit et al., 2002). OIB trace element pattern for sample SA11 from the ITE-rich series (Fig. 4) support the asthenospheric mantle involvement. On Sr-Nd isotopic plot this samples deviates further toward and plot within the mantle array as compared to the other samples from the ITE-rich series.

### 7.5 Slab/crustal contributions to the arc magmatism

    As differentiation by fractional crystallization (FC; see above) proceeds in a magmatic series, the more evolved melts show

higher abundances of incompatible trace elements (e.g., Zheng, 2019). This is because the fractionated mineral assemblage has low Kd for the incompatible trace elements. However, in some batches of differentiated samples the spate of increasing abundances of a few incompatible trace elements such as Th, K, La, and Ce exceeds that of the other incompatible trace





elements. These are believed to indicate crustal contamination or assimilation which is usually associated with FC (i.e., assimilation fractional crystallization, AFC). Arc mafic volcanic rocks, due to the rather high ascent rate of mafic melts, are

less likely to have undergone significant continental crustal contamination. Hence, their crustal signatures have probably been derived from the slab (i.e., slab plus sediment) fluids and melts that metasomatized the overlying mantle wedge (e.g., Benoit et al., 2002).

Origin and evolution of mafic volcanic rocks are best represented by the Sr-Nd-Pb isotopic ratios. The three series of volcanic rocks from the UDMA (i.e., the LILE-poor, LILE-rich, and ITE-rich series) plot on the same area of the Sr-Nd

isotopic plot as other orogenic volcanic rocks and on the same area as other mafic volcanic rocks from the UDMA (Fig. 5). However, subtle but significant differences are observed in distribution of the three series on the Sr-Nd isotopic plot. Toward lower Sr isotopic ratios, data points from the three series converge to share a narrow space in the mantle array close to the BSE. The Sr-Nd isotopic variation trends expand towards higher Sr and lower Nd isotopic ratios (Fig. 5.a). Increasing Sr isotopic ratios occurs at constant Nd isotopic ratios in the LILE-rich series whereas in the LILE-poor and ITE-rich series, Sr

isotopic ratios increase at decreasing Nd isotopic ratios (Fig. 5.a).

On Pb-Pb isotopic variation diagrams (Fig. 5.b, c) the least radiogenic members of the three series of mafic volcanic rocks from the study areas, the LILE-poor, LILE-rich and ITE-rich series, plot on the DMM-OIB mantle array (i.e., on modestly depleted mantle area of the NHRL). The more differentiated volcanic rocks demonstrate sharp enrichment trends towards radiogenic Pb end-members on the Pb-Pb isotopic plots. Interestingly however the three series of volcanic rocks share one

variation trend on these plots; that means the wet-slab-melt and dry-slab-melt (i.e., responsible for development of the LILE-rich and LILE-poor series respectively) maintain the same Pb isotopic ratios. This implies that unlike different LILE/MREE ratios in the slab components involved in petrogenesis of the LILE-rich and LILE-poor series, these slab-derived components have the same U and U/Th.

It was noted above that geochemistry do not support continental crustal contamination (see Fig. 6). The isotopes are not fully

discriminant to support the trace element classification of mafic volcanic rocks into the three series presented here. However, the isotopes might be utilized to evaluate/estimate the geochemical reservoirs involved in their petrogenesis. Assuming a negligible continental crustal contamination in evolution of the UDMA mafic volcanic rocks and utilizing Sr-Nd isotopic data, a ternary mixing modeling is carried out that shows the UDMA mafic volcanic rocks can be best explained by the mantle wedge and slab (i.e., including slab melt and sediment melt) participation.

The mantle wedge, slab and slab sediment are respectively represented by the average of Kamchatkan arc volcanic rocks (Kepezhinskas et al., 1997), an altered oceanic crust (AOC; Hauff et al., 2003), and the Tethyan flysch sandstone sample 06FL03 from Serbia (Prelević et al., 2008). The ternary mixing model applied for the UDMA mafic rocks found that in the LILE-rich and LILE-poor series, mantle wedge:slab contributions were 45:55 and 55:45 respectively. Numbers marked on the red and blue mixing curves in the inset of Fig. 5a indicate that slab involvement in petrogenesis of the LILE-rich and

LILE-poor series were 50-60% and 40-50% respectively. For simplicity these are presented as mean value of 55 and 45%.





The yellow curve is not recommended as a mixing model for development of the ITE-rich series. As mentioned earlier, the ITE-rich series has likely been generated in a process similar to the one that led to the high-Nb basalts (Section 7.5).

Intercept of the red and blue mixing curves with the slab melt-slab sediment array marks the proportion of slab melt:sediment melt involved in petrogenesis of the LILE-rich and LILE-poor series (i.e., 50:50 and 20:80) respectively.
Combining these two figures, the mantle wedge:slab and the slab melt:sediment melt proportions, the important mantle wedge:slab melt:sediment melt proportions of 45:27.5:27.5 and 55:09:36 are obtained for the LILE-rich and LILE-poor series respectively. Mafic volcanic rocks from the W Nain area of the central UDMA (Yeganehfar et al., 2013) with trace element patterns that mimic the LILE-rich, LILE-poor and ITE-rich series patterns (Fig. S4) show Sr-Nd-Pb isotopic characteristics that are similar to the respective series from the study area further supporting the petrogenetic model
presented here (Fig. A5).

The least evolved volcanic rocks are presented in this study, however rocks from the Nodoushan (i.e., southeastern segment of the study area) are still highly differentiated (see Section 7.3 and Fig. S2.b). The Nodoushan rocks also show significant crustal contamination signatures such as elevated lithophiles, Th, Pb, La-Ce and Sr isotopic ratios. Some of the rocks show the highest Sr isotopic ratios from the study area. Hence, they are not included in the isotopic modelling. One of the
Nodoushan rocks show the highest Nd isotopic ratios, indicating the likely involvement of a more depleted mantle source.

**7.6 Mantle heterogeneity revealed by the isotopes**

On the Sr-Nd isotopic plot, two samples from the LILE-poor series and one sample from the ITE-rich series are offset (i.e., from the respective series) towards significantly more radiogenic and less radiogenic compositions respectively. Given $SiO_2$ as differentiation index, the two samples from the LILE-poor series are the least differentiated samples in the series (Fig. 6;
$SiO_2$ vs Nd and Sr isotopic ratios) yet they show the highest Sr isotopic ratios. Therefore, the isotopic signatures of these two sample from the LILE-poor series are less likely to be the consequence of higher degree of contamination by slab and sediment melts (i.e., in the same metasomatism event). Particularly so because of the rather low Rb contents of these two samples; sample UE41 with the highest Sr isotopic ratio contain the least Rb abundances (Tables S2 and S1). The older Nd model age of these two samples from the LILE-poor series as compared to the other mafic rocks from the respective series,
averaging 0.90 Ga and 0.82 Ga respectively (Fig. 8), imply the compositional heterogeneity imposed on the mantle by earlier metasomatic events.



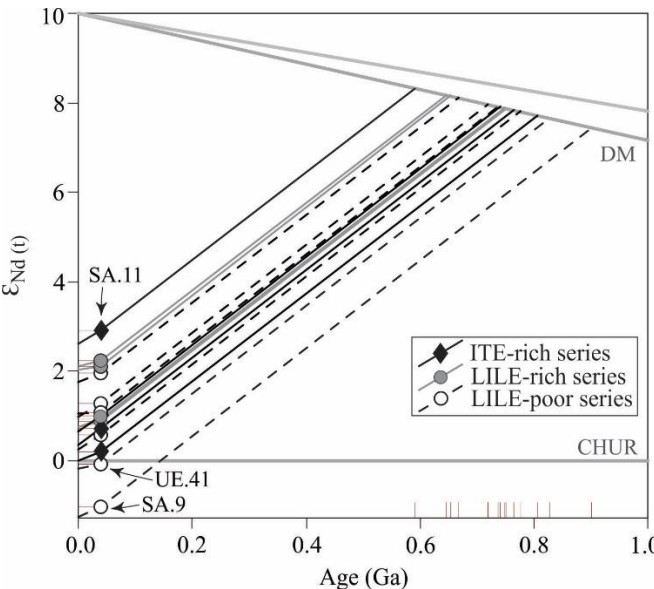

**Figure 8: Age (Ga) vs. εNd(t) values plot for the LILE-rich, LILE-poor, and ITE-rich series mafic volcanic rocks from the central**
**UDMA. Abbreviation: CHUR, chondritic uniform reservoir (DePaolo and Wasserburg, 1976); DM, depleted mantle.**

**8 Conclusions (Temporal evolution of the UDMA volcanic rock series and their implications)**

Trace element and Sr-Nd-Pb isotopic geochemistry of the mafic volcanic rocks from the Urumieh-Dokhtar Magmatic Arc (UDMA) of Iran indicate a metasomatized mantle origin and suggest a ternary classification, into the LILE-rich, LILE-poor
and ITE-rich series. This is interpreted to suggest the nature of metasomatizing agents involved (i.e., the wet-slab-melt, the dry-slab-melt and more fertile segments of the mantle) in the development of the respective series. A scrutiny on the ages of the volcanic rocks of the three series compositional affinities (i.e., the LILE-rich, LILE-poor, and ITE-rich series) from the UDMA indicates Eocene to Early Miocene ages for these three series. This favors an intermittent volcanism for these rock series over an extended period of time rather than an episodic magmatism of a particular type. It is suggested here that the
mantle metasomatism responsible for the UDMA magmatism predates the Eocene time during which the metasomatized mantle reservoirs of the LILE-rich and LILE-poor have been developed (i.e., by wet-slab-melt and dry-slab-melt respectively) in the mantle. Similar Sr-Nd-Pb isotopic compositions to the other contemporary arc-type UDMA mafic rocks suggest a source modified by recycled slab (slab + sediment). Isotopic modelling indicates that mafic UDMA volcanic rocks from the LILE-rich and LILE-poor series are derived from juvenile metasomatized mantles with mantle:slab contribution of
45:55 and 55:45 respectively wherein the slab melt:sediment melt involvements were 50:50 and 20:80. The ITE-rich series bear the signatures of source mantle metasomatism by slab melts prompted by and interacted with asthenospheric melts in a slab window. Rather high Nd model age and higher Sr isotopic ratios of the less differentiated mafic volcanic rocks from the



LILE-poor series represent older metasomatic events hence indicating source mantle heterogeneity. In the Eocene-Miocene interval, depending on the thermal gradient and tectonic regime at the time, these metasomatized mantle reservoirs have
intermittently undergone partial melting and produced the LILE-rich, LILE-poor, and ITE-rich series rocks.

*Supplementary data.* Supplementary material includes Tables S1–S3 and Figs. S1–S5.

*Data availability.* Data are available in EarthChem Library and can be accessed via a DOI link at https://doi.org/10.60520/IEDA/113464.

*Author contribution.* **MRG:** Conceptualization, Data curation, Formal analysis, Funding acquisition, Investigation,
Methodology, Project administration, Resources, Supervision, Validation, Visualization, Writing – original draft preparation, Writing – review & editing. **MA:** Conceptualization, Data curation, Formal analysis, Investigation, Methodology, Project administration, Resources, Software, Supervision, Validation, Visualization, Writing – original draft preparation, Writing – review & editing. **ITG:** Conceptualization, Data curation, Formal analysis, Funding acquisition, Resources, Software, Validation, Visualization, Writing – review & editing. **MB:** Conceptualization, Data curation, Formal analysis, Funding
acquisition, Investigation, Methodology, Resources, Software, Validation, Visualization, Writing – review & editing. **FS:** Conceptualization, Data curation, Formal analysis, Validation, Visualization, Writing – review & editing.

*Competing interests.* The authors declare that they have no conflict of interest.

*Acknowledgments.* This work is a part of the sabbatical leave that the first author spent at the University of New South Wales (UNSW, Sydney, Australia) in early 2023 as well as a part of his long-term research interest on the arc magmatism in the
Tarbiat Modares University (TMU, Iran). Funds and facilities provided by the UNSW, GET-OMP (Toulouse, France), and TMU are acknowledged.

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
