# Peer review of "What do arc magmatism trace element patterns and Sr-Nd-Pb isotopic data reflect: Insights from the Urumieh-Dokhtar magmatic arc of Iran"

_EGUsphere, 2024_

## Author Response (AR1)

**Response to Reviewers**

(point-by-point)

Dear Prof. Coltorti,

Firstly, we would like to thank you for your handling of our manuscript and for the valuable comments made by the referees. Detailed below are our comments and revisions to the manuscript based on the recommendations and comments made by the referees.

**Reviewer #1 (Dr. Pier Paolo Giacomoni)**

**Comment 1:** The "geology of the study area" section is too crowded of references, often too scholarly. I suggest reducing the number of references, those referring to the fig.1 should be included in the caption.

**Response:** As suggested by Reviewer #1 (Dr. Pier Paolo Giacomoni), the number of the references in the "Geology of the study area" section has been reduced. Those referred to in the Fig. 1, are now included in the figure caption.

**Comment 2:** Petrography

The "petrography" section is a little bit confused, maybe could be re-organized. I suggest adding a table which summarize the main petrographic features of the studied basic samples such as: porphyritic index, index of vesicles, mineralogic assemblage and relative proportions of phenocrysts and other petrographic additional features. Micro photos are not useful in the presented form. I suggest to carefully select few representative low magnification images and to add them in the table.

**Response:** As suggested by the reviewer, the main petrographic characteristics of the mafic rocks from the study area have now been added to the manuscript as Table 1. A representative set of low-magnification petrographic images have also been added to the manuscript as Fig. S2; these figures are referred to in Table 1.

**Comment 3:** Major and trace element analysis

Please provide more info about standards nature and quality for both XRF and ICP-MS. How did you consider the $Fe^{2+}/Fe^{3+}$ in the melt? Did you correct the Fe partitioning after loss on ignition (LOI) gravimetric measurements?

**Response:** Both in-house and internationally certified standards measured using the ICP-MS method at the Genalysis Laboratory and the University of Melbourne are now provided in Table S4. Due correction is made in the last paragraph of section "Major and trace element analysis". In the present study, the Fe content is presented as $FeO^t$. Due correction is made in the manuscript;

FeO$^t$ replaced earlier expressions (FeO, Fe$_2$O$_3$). No recalculation to Fe$^{2+}$ and Fe$^{3+}$ has been carried out.

**Comment 4:** Results

In fig.3 the diagram shows the Fe$_2$O$_3$ variation while is described as FeO in the text. It is very important to clarify how do you calculate the FeO/Fe$_2$O$_3$ ratio and keeping consistency in the result section and discussion.

I agree with most of the thesis treated in the discussion section however, I suggest to also plot a Mg# versus CaO/Al$_2$O$_3$ in order to evaluate both primitivity of melts and their sensitivity to Ol+Cpx+Pl fractionation. Observations in the 7.3 section are valid however should be supported by a numerical fractionation modelling.

**Response:** As suggested by Reviewer #1, and in being consistent, all references to the iron content of the samples are now referred as FeO$^t$ throughout the manuscript.

Differentiation trends furnished by the MagPath spreadsheet program (Mayborn and Lesher, 2011) are found as more rigorous tools in representing magmatic evolution than the Mg# versus CaO/Al$_2$O$_3$ plot, so the former are presented on the Harker variation diagrams (Fig. 2). The CaO/Al$_2$O$_3$ versus SiO$_2$ plot is also added in Fig. 2.

Adding fractional crystallization modeling to the Harker diagrams entails adding a note to the Figure caption as well as adding the following note to the second paragraph of section 6.3 (Does fractional crystallization provide an interseries link?):

"5- Fractional crystallization modelling using the MagPath algorithm (Mayborn and Lesher, 2011) demonstrates different liquid lines of descent for the LILE-rich and LILE-poor series developed as a result of differentiation of distinct mineral assemblages (see Fig. 2 caption for details)."

**Comment 5:** Line 361

Line 361. "In contrast, the microcrystalline groundmass of the samples from the LILE-poor series is due to the higher viscosity of their primitive melts as a consequence of their mantle source metasomatism by silica-rich slab partial melts"

In diagrams in fig. 3 there is no significant SiO$_2$ content variation between LILE-rich and LILE-poor. The viscosity is a consequence of the matrix crystallinity not vice-versa. Sample matrix crystallinity is more likely a consequence of cooling rate and volatile exsolution (undercooling) before and during eruption, thus not depending of mantle source features. This is especially true considering that studied magmas are not representative of primary melts but underwent to some degree of fractionation en-route to the surface.

I suggest to re-arrange this reasoning by considering the differing contribution of hydrated vs dehydrated slab in adding water and other volatiles to primary melts. More hydrated melts tend to crystallize later, reducing the stability of plagioclase while enlarging that of pyroxene. Hydrated magmas should result in less crystalline groundmass with few plagioclase phenocrysts.

**Response:** As implied by the petrographic characteristics presented in the "new" Table 1, the LILE-rich and LILE-poor series volcanic rocks from the study area show widely overlapping textural and mineralogical characteristics. However, geochemical variation of the volcanic rocks coupled with the inferred mineralogical phase relations help project some petrogenetic implications. As a consequence of reevaluation of the petrography, the following text (in section 6.4; second paragraph) has been changed:

"responsible for its lower melt viscosity that is reflected in the aphyric groundmass of samples from this series (Fig. 2). In contrast, the microcrystalline groundmass of the samples from the LILE-poor series is due to the higher viscosity of their primitive melts as a consequence of their mantle source metasomatism by silica-rich slab partial melts."

is replaced by:

" consistent with geochemical evolution of the LILE-rich series; that is the hydrous nature of the LILE-rich primitive melt depressed plagioclase fractionation thereby prompted an early Al increase in the 48-52 wt% silica range in the series."

**Reviewer #2 (Prof. Delphine Bosch)**

*Revisions made in regard with the Reviewer #2 comments:*

**The 1ˢᵗ bullet point:** i- provide, more consistently, evidence that the trace element balance of the different samples was not significantly affected by crystallisation fractionation processes. In other words, that the different spectra observed do not result from a co-precipitation effect of the minerals. This can be achieved by drawing new figures combining certain LILE elements according to a differentiation index such as $SiO_2$ wt% or MgO wt%. This has major consequences for the author's interpretations, because if the distribution of trace elements can be proven to be linked to fractionation processes, then the different batches of samples proposed by the authors are not strongly demonstrated to be effective, nor is their origin ;

**Response:** This reviewer has suggested adding some plots that include LILE, in order to demonstrate the independence of the major rock series (i.e., namely the LILE-rich and LILE-poor series). A proof that one of these series is not developed by the differentiation of the other series. To carry out this task, the $SiO_2$ versus Ba/Nb and $SiO_2$ versus Ba/La plots are now added to the manuscript (Fig. 6.a, b). On these plots the more mafic samples show distinct compositional spectra that support different origins for the LILE-rich and LILE-poor series.

The following text is now added to the 3ʳᵈ paragraph of section 6.3 (Does fractional crystallization provide an interseries link?):

"Further assurance as to the primitive nature of the ternary classification of the volcanic rocks from the study area (i.e., that the geochemical distinction between these three rocks series are not an artifact of fractional crystallization) are provided by a series of plots where the mafic samples define distinct groupings/series. $SiO_2$ versus Ba/Nb or Ba/La help distinguish the LILE-rich from

the LILE-poor series (Fig. 6.a, b) whereas $SiO_2$ versus Nb or Zr help distinguish the ITE-rich series from the other two series (Fig. 6.c, d)."

**The 2ⁿᵈ bullet-point comment:** better demonstrate the validity of the proposed subdivision into three categories of samples, namely ITE-rich samples, LILE-poor samples and LILE-rich samples. Looking at Figure 4, the distinction between LILE-poor and LILE-rich samples is not obvious, even for Kahak or Ardestan samples. Please add additional figures using individual LILE and inter-element variations to unequivocally demonstrate the existence of the three batches of samples proposed on the basis of strong discriminating features;

**Response:** The reviewer requested a better demonstration of the validity of the subdivision of samples from this study into the three categories; the LILE-rich, LILE-poor, and ITE-rich series. By adding the above mentioned plots; $SiO_2$ versus Ba/Nb, Ba/La, Nb, and Zr, this has now been carried out (see Fig. 6).

**The 3ʳᵈ bullet-point comment:** the discussion of fluids and melts (wet or dry) from the subducted slab merit further development as, as it stands, the authors first suggest the contribution of wet slab melts and dry slab melts to explain the LILE-rich and LILE-poor series, respectively. However, later in the text, they also suggest the involvement of fluids. To help the reader follow the proposed scenario, it would be easier to present the different processes in the order in which they occur, i.e. dehydration first, then wet slab melting and eventually dry slab melting. Fluid or melt input can also be controlled using ratios of elements with similar partition coefficients during mantle melting but contrasting behaviour in aqueous fluids [e.g. Ba/La, U/Th and Pb/Ce]. In the discussion, the distinction between the contribution of fluids and the contribution of melts is not clear. Th is an element considered to be immobile, but if there is a melting process, even a wet one, it will be enriched in the corresponding rock. The conditions required to produce melting in a subduction zone impose strong changes in the subduction parameters and in particular a higher plate temperature of the order of 700-900°C at depths less than the arc, conditions required to transfer significant quantities of trace elements from the plate to the mantle (see Hermann and Spandler, 2008; Hermann et al., 2006 and others);

**Response:** The term wet-slab-melt and dry-slab-melt have now been modified into fluid-rich slab melt and fluid-poor slab melt to better match the realities. The text has also been rechecked to make sure that the term "fluid-involvement" communicates the degree of contribution by one of these two end-members (i.e., fluid-rich slab melt and fluid-poor slab melt).

As per slab melting conditions, the following revision has been made. This includes an emphasis on the spatial distinctions of the LILE-rich and the LILE-poor series that furnishes further explanations as to their distinct petrogenesis.

In section 6.4.2 (i.e., Simultaneous fluid-rich slab melts and fluid-poor slab melts mantle metasomatism scenario) the following text:

"At first, hydrous slab melting at a given depth lead to the mantle metasomatism by hydrous slab partial melts. It is shortly followed by a modest change in the physiochemical conditions that governed the slab-mantle wedge system. The new conditions prompted partial melting of the slab

that has already undergone dehydration, leading to the development of second type of mantle metasomatism by anhydrous slab partial melts"

has been replaced by the following text:

"Young hot slabs are capable of yielding partial melts at subarc depths that correspond to the volcanic front (Hermann and Spandler, 2008; Manea et al., 2014; Zheng, 2019). Hot slab subduction of then narrow/almost vanished Neotethyan oceanic plate has likely been responsible for petrogenesis of the slab melts that metasomatized the source mantles of the three series; the LILE-rich, LILE-poor and ITE-rich series. Disappearance and termination of the Neotethyan oceanic slab is estimated to have predated the early Cenozoic (Agard et al., 2005; Horton et al., 2008; Dargahi et al., 2010). The LILE-rich and LILE-poor series rocks are mainly spatially constrained towards the northern and southern parts of the study area. This implies a more hydrated, and more altered nature of the northern segment of the slab whereas the southern segment of the slab is deemed to have been of higher T (i.e. they might have been thicker and thinner slabs respectively)."

**The 4th bullet-point comment:** the ternary mixing model used to calculate the different proportions of the various end-members involved in the mixing lacks details on the procedure used to obtain such proportions such as the formulae, what type of mixing? mass to mass?, KD, the errors in the percentages obtained?;

**Response:** In the revised Table S5, we have now included a more comprehensive explanation of the procedure used to calculate the proportions of the different end-members. Details have also been added to the modified Table:

"The mixing proportions were determined using a simple mass-balance approach, adapted from the two-end-member mixing equations proposed by Powell (1984). For each element, the concentration in the mixed magma ($C_m$) can be expressed as:

$$C_m = X\,(C_a - C_b) + C_b$$

Where $C_a$ and $C_b$ are the concentrations of the element in the two end-member magmas, and $X$ represents the degree of mixing. For isotopic compositions, the equation is:

$$I_{C_m} = I_{C_a}\left(\frac{C_a X}{C_m}\right) + I_{C_b}\left(\frac{C_b\,(1 - X)}{C_m}\right)$$

where $I_{C_a}$ and $I_{C_b}$ are the isotopic ratios in the two end-member magmas, and $I_{C_m}$ is the isotopic ratio in the mixed magma.

**The 5th bullet-point comment:** section 7.6 is unclear as it stands. I would therefore recommend to remove it.

**Response:** As suggested by the Reviewer #2, Section 7.6 that include a figure (i.e., Age vs. $\varepsilon$Nd) is removed. As a result, the following modifications have been made:

In Fig. 5: The following text is added in the end of the figure caption:

"Gray areas/bands marks restricted Sr-Nd isotopic variation for most of the mafic rocks. One sample from the ITE-rich series plots towards less radiogenic Sr-Nd isotopic ratios (see Section 6.5 and Fig. 4.a for details). Two samples from the LILE-poor series plot towards more radiogenic Sr-Nd isotopic ratios; this is due to the higher slab components (i.e., slab + sediment) involvement (see Section 6.6 and Fig. 4.a)."

The related text is removed from Abstract.

The related text is removed from Conclusion.

In Section 6.3 (Does fractional crystallization provide an interseries link?), second paragraph, the phrase:

"….. see Sections 7.5 and 8)"
is replaced by:
"….. see Sections 6.5 and 6.6 and Fig. 4)"

In Section 6.5 (The incompatible trace elements-rich (ITE-rich) series), the following is added at the very end of second paragraph:

"(see caption in Fig. 5)"

*Miscellaneous*

**Comment:** l.145: "Accessory minerals are trace element repositories that resist the mixed acid digest method." This is true for zircon or spinel but not for other accessory minerals.

**Revision made:** The term "Accessory minerals" is now replaced by "Some Accessory minerals".

**Comment:** Section 3 "$E_1$-$E_6$ subdivisions versus age dated volcanic succession »: in its present state, I think this paragraph fits better in the Geological background section.

**Revision made:** As suggested by the referee, the content of this section is now distributed in three parts of the section "Geology of the study area".

**Comment:** Section 5.2: please indicate the error level for Nd isotopes. This data is also missing in Table S2. Please fill in.

**Revision made:** The error level for Nd isotopes is now added to Table S3.

**Comment:** l.155: precision on U content is mentioned but this element is not indicated in Table S1 nor in the trace elements patterns. Why?

**Revision made:** U is now added in the trace element patterns (Fig. 3) and in Table S2. The following text is also added as the 3rd paragraph in Section 6.2 (Primitive signatures):

"Most of the mafic volcanic rocks from the study area show deep negative anomalies for Th but mild or no anomaly for U similar to the island arc volcanic (Hawkesworth et al., 1997). It is because Th behaves like HFSE whereas U is mobilised by fluids. However, a few samples from

the LILE-poor series show positive Th-U anomalies (Fig. 3). One of these samples show the highest LREE and P abundances amongst the mafic rocks (sample SA.22; Fig. 3). These might indicate apatite involvement in petrogenesis of the mafic rocks with positive Th-U anomalies. Apatite show extremely high partition coefficients for Th, U, and LREE (O'Reilly et al., 1991). Morishita et al. (2003) found that apatite development in the Finero peridotite is due to the metasomatizing agent derived from subducting 'slab possibly containing small quantity of sediment', the condition supported by the isotopic modelling of source end-members for the mafic volcanic rocks from the study area (see Section 6.6)."

**Comment:** Section 7.3.2 "Simultaneous slab fluids and slab melts mantle metasomatism scenario": I do not follow how it will be possible to produce simultaneously dehydration and melting processes at a same place?

**Revision made:** Due corrections are made in this section of the manuscript. These include the following:

"The LILE-rich and LILE-poor series rocks are mainly spatially constrained towards the northern and southern parts of the study areas, namely the Kahak and Ardestan areas respectively."

Thereby it is made clear that different styles (compositions/affinities) of magmatism occurred concurrently in different parts of the study area.

The above mentioned revisions made in reply to the reviewer comments are also reflected in the Abstract and Conclusion sections as shown in the attached "Track changed" file.